# A Tree-Based Heuristic for the One-Dimensional Cutting Stock Problem Optimization Using Leftovers

**DOI:** 10.3390/ma16227133

**Published:** 2023-11-11

**Authors:** Glaucia Maria Bressan, Matheus Henrique Pimenta-Zanon, Fabio Sakuray

**Affiliations:** 1Mathematics Department, Universidade Tecnológica Federal do Paraná (UTFPR), Alberto Carazzai, 1640, Cornélio Procópio 86300-000, PR, Brazil; 2Computer Science Department, Universidade Tecnológica Federal do Paraná (UTFPR), Alberto Carazzai, 1640, Cornélio Procópio 86300-000, PR, Brazil; matheus.pimenta@outlook.com; 3Computer Science Department, State University of Londrina (UEL), Rodovia Celso Garcia Cid, Pr 445 Km 380 C.P. 10.011, Londrina 86057-970, PR, Brazil; sakuray@uel.br

**Keywords:** heuristic procedures, usable leftovers, cutting problems, optimization

## Abstract

Cutting problems consist of cutting a set of objects available in stock in order to produce the desired items in specified quantities and sizes. The cutting process can generate leftovers (which can be reused in the case of new demand) or losses (which are discarded). This paper presents a tree-based heuristic method for minimizing the number of cut bars in the one-dimensional cutting process, satisfying the item demand in an unlimited bar quantity of just one type. The results of simulations are compared with the $RGR_{L1}$ algorithm and with the limiting values for this considered type of problem. The results show that the proposed heuristic reduces processing time and the number of bars needed in the cutting process, while it provides a larger leftover (by grouping losses) for the one-dimensional cutting stock problem. The heuristic contributes to reduction in raw materials or manufacturing costs in industrial processes.

## 1. Introduction

The cutting stock problem (CSP) consists of cutting a set of objects available in stock to produce the desired items in specified quantities and sizes in order to optimize (maximize or minimize) a given objective function [1] such as production costs or losses from the cutting process.

Many industry processes are using the CSP, such as the automotive, construction and bicycle manufacturing industries. Considering 16, 16 and 14 m bars in stock and 3, 4 and 6 m items, with demands of 2, 3 and 2, respectively, as in Figure 1. In this process, the loss and leftover are classified by size of the bar not used, i.e., a bar smaller than a smallest item demanded is considered loss or is otherwise leftover.

Figure 2 shows two possible solutions: in Figure 2a, only the 16 m bars are used, with 2 m of loss. In Figure 2b, one 16 m bar and one 14 m bar are used, without loss. In both patterns, leftovers are produced, but in Figure 2b, there is no loss.

The CSP belongs to an important area of operational research that assists in the decision-making process. There are different types of mathematical models that can formulate the cutting stock problems. One of the possible models is linear programming, which proposes to optimize the objective function, subject to a set of constraints, whose variables are linearly related [1]. The one-dimensional cutting stock problem in turn involves only one dimension in the cutting process, such as cutting steel bars, paper rolls and tubes [2,3].

Gilmore and Gomory were among the first to address the cutting stock problem [4,5] and to define the problem as filling an order at minimum cost for specified lengths of material to be cut from given stock lengths of given cost. Also, in [6], higher-dimensional cutting stock problems are modeled as linear programming problems.

The *usable leftover cutting stock problems* [2] consist of a branch of the cutting stock problem. Its purpose is to determine the cutting patterns and to analyze the leftovers generated by the cutting process. This type of problem can be understood as a one-dimensional cutting stock problem in which the unused material in the cutting process may be used in the future, if large enough [7]. A review of published studies that consider the solution of the one-dimensional cutting stock problem with the possibility of using leftovers to meet future demands is presented by [3].

A heuristic algorithm for the one-dimensional cutting stock problem with usable leftovers is proposed in [8], which consists of two procedures: linear programming that fulfills the major portion of the item demand and a sequential heuristic that fulfills the remaining portion of the item demand. In [2], it is assumed that leftovers should not remain in stock for a long time. Thus, leftovers have priority-in-use compared to standard objects (objects bought by the industry) in stock, and a heuristic procedure is proposed for this problem.

Addressing more recent papers, Cui et al. [9] present an integer programming model for the one-dimensional cutting stock problem with limited leftover types and describe a heuristic algorithm based on a column-generation procedure to solve it. The column generation technique is used in [10], and it is associated with a set of residual cutting patterns. It recombines these residual cutting patterns in different ways and generates new integer-feasible cutting patterns. In [11], a modification in the constructive greedy heuristic, which builds a cutting pattern by sorting in descending order the paired items or items of odd length, is presented. The only objective is to minimize the quantity of cut objects.

Considering the cutting stock problems and usable leftovers, the objective of this paper is to propose a tree-based heuristic method, named *OptimizationTREE*, in order to minimize the number of bars needed to cut known item demand, in the one-dimensional cutting process with no stock. The problem presented in this paper considers an unlimited quantity of one type of bar.

These methods are measured in terms of accuracy and runtime. The proposed heuristic method is described in the next section. Its performance was analyzed and compared with a well-known algorithm in the literature: the $RGR_{L1}$ algorithm, proposed by [7]. The accuracy was rated according to: (i) total number of used bars, (ii) the loss and leftover produced in the process. The runtime shows that the algorithms present better computational complexity. The proposed heuristics aim to reduce losses by grouping multiple losses in the same bar (whenever possible), so that the losses (useless) become leftovers (useful), as shown in Figure 2.

The main contribution is to obtain optimized solutions, similar to those obtained in [7], for the one-dimensional cutting stock problem, with relevant gains in computational complexity (or runtime) and conversion of loss to leftovers whenever possible. The heuristics can be used in commercial software solutions and provide a reduction in the raw materials used (or manufacturing costs) and, consequently, environment preservation, as was the case for the results obtained by [12,13], which address the green manufacturing study topic.

The novelties of this proposal, in relation to existing methods in the literature, are highlighted as follows: (i) the heuristic is based on a tree structure, which provides accessibility and understanding for users, in addition to a reduction in processing time; (ii) multiple losses are grouped on the same bar whenever possible, transforming losses into reusable leftovers and providing loss reduction. The algorithm does not require extra raw material; that is, it uses the same quantity of initial material for cutting.

The remainder of the paper is organized as follows. Section 2 presents the definition of the problem under study in this paper. Section 3 describes the initial procedure used in the first step of the proposed heuristic, whose algorithm, named *OptimizationTREE*, is presented. The data set for the simulation is provided in Section 4, followed by analysis of the simulation’s results (Section 5). Finally, in Section 6, the conclusions of the paper are given, and aspects of future work are discussed.

## 2. Definition of the Problem under Study

The proposed heuristic methods aim to present solutions for the general problem defined as follows. During the cutting process, unavoidable leftovers are produced and often discarded. Industries have the possibility of using the leftover to cut future demanded items, since their sizes are sufficiently large, that is, the minimum acceptable length to be reused in the case of new demand, which is a choice of the decision maker [7]. In this situation, the simple objective of minimizing the leftover may not be appropriate.

Problem: To minimize the number of bars needed to cut known item demand, in the one-dimensional cutting process, the demand of items has to be met by cutting bars of the same size:(1)$$\sum_{p=1}^{P}x_{ip}bar_{p}\geq d_{i},\forall i\in 1,\ldots ,n,bar_{p}\geq 0\;\mathrm{and}\;\mathrm{integers},$$

The variables and parameters are described below.

$bar_{p}$: number of cut bars in the cutting patterns p=1,…,P (variable);$d_{i}$: demand of item *i*;$x_{ip}$: the number of items i=1,…,n cut in the cutting pattern *p*. In the one-dimensional case, any cutting pattern must satisfy [7].
(2)$$\sum_{i=1}^{n}l_{i}x_{ip}\leq L,\forall p\in 1,\ldots ,P;0\leq x_{ip}\leq d_{i},$$where *L* is the size of the bar, and $l_{i}$ is the length of item *i*. This expression guarantees that the number of produced items is not larger than the size of the bar.

Thus, the cutting process can generate leftovers (which can be reused in the case of new demand) or losses (which are discarded). The goal is to reduce losses by grouping them in the same bar when possible. Thus, the losses become leftovers and can be used in future demand.

## 3. The Proposed Heuristic: *OptimizationTREE*

In this section, the proposed heuristic method to solve the problem described in Section 2 is presented. The heuristic was evaluated in terms of the total number of bars used in the whole cutting process, the loss and leftover provided and also the computational complexity of the solution method.

First of all, a selection of ordered items (SOI) procedure, described in Algorithm 1, must be presented, since this ordering procedure, which is based on the first fit algorithm [14], is used in the first step of the proposed heuristics.

This procedure uses as input the set of items demanded (**l** and **d**), sorted in descending order of size. Then, the items are selected to be cut, starting with the largest one; if this is not possible, the next item is tested and so on. Other inputs are the size of the bar (*L*) and the total number of items (*n*). The outputs are the number of each item *i* in the cutting pattern or bar (**x**), the size of the bar not used ($\hat{L}$) and the updated demand of items (**d**).
**Algorithm 1:** Selection of ordered items (SOI)
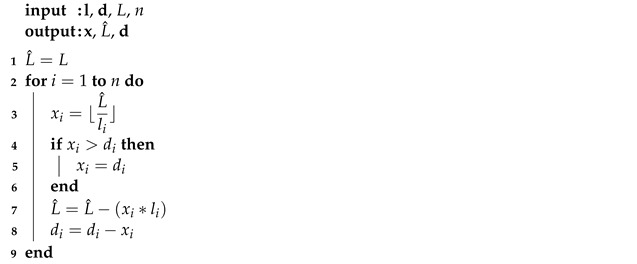


    A tree-based heuristic (*OptimizationTREE*) is described in Algorithm 2. It has two phases: (i) selection of the items to be cut and (ii) loss reduction. First, the set of demanded items is sorted in descending order of size (**l**). In the first phase, a set of items is selected according to Algorithm 1. This first set is then used in the second phase to reduce losses as follows: for each selected item, the number of items to be cut is reduced by one unit in an attempt to fill the rest of the bar with larger elements and then with smaller elements. After the next smaller item to be cut is selected, the algorithm verifies whether a larger item can be used; if not, the next smaller item is processed. The inputs, outputs and main variables of Algorithm 2 are described as follows.

The inputs of Algorithm 2 are:The item size $l=\{l_{1},\cdots ,l_{n}\}$;The item demand $d=\{d_{1},\cdots ,d_{n}\}$;The size of bar *L*;The number of items to be cut *n*.

The outputs of Algorithm 2 are:The leftover of all cutting processes;The loss, the sum of loss in each bar used;The bar with the sum of used bars.

The variables used in Algorithm 2 are described as follows:$\hat{L}$: size of the bar not used during the execution of the algorithm;$x_{i}$: the number of items *i* to be cut in the bar, for i=1,⋯,n.

In this proposed heuristic, the losses of the cutting process are concentrated on the smallest number of bars possible, using a tree structure [15], in order to convert losses (unusable) into leftovers (usable) and reduce the number of bars needed in the process.
**Algorithm 2:** Tree-based heuristic (*OptimizationTREE*)
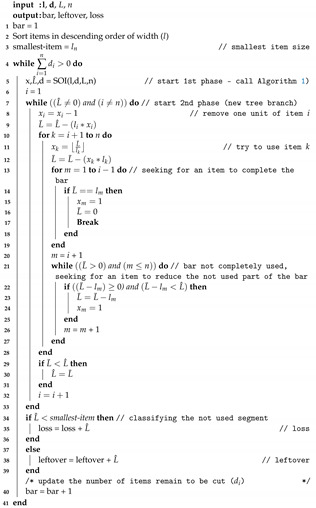


    A hypothetical optimal solution can be determined as a parameter to obtain the minimum number of bars required to produce the demanded items. This theoretical minimum number of bars needed to produce the demanded items is defined in Equation (Equation 3). The *lower bound of bars* ($B_{bar}$) is used to evaluate the performance of the proposed heuristics, i.e., better solutions use bars close to $B_{bar}$. If any of the heuristics provide as a solution a minimum number of cutting bars smaller than $B_{bar}$, it means that the algorithm has an error, and the solution is impossible.
(3)$$B_{bar}=\left\lceil \frac{\sum_{i=1}^{n}l_{i}d_{i}}{L}\right\rceil$$

When this theoretical limit is achieved, the unused material remaining is defined in Equation (Equation 4).
(4)$$\begin{array}{c} Minimum\;Theoretical\\ limit\;of\\ Unused\;Material \end{array}=\left(B_{bar}-\frac{\sum_{i=1}^{n}l_{i}d_{i}}{L}\right)L$$

The minimum theoretical limit of unused material (Equation (Equation 4)) can be classified as follows:Loss ($B_{loss}$): if the unused material is smaller than the smallest demanded item;Leftover ($B_{left}$): if the unused material is greater than or equal to the smallest demanded item.

## 4. Data Set for Simulations

The numerical values used in simulations of the proposed heuristics are based on [16], considering a one-dimensional cutting stock problem for steel roll. In this paper, a *category* is a set of $d_{i}$ (i=1,…,n) and *n* (the number of items). In the simulations of this study, 10 categories, as shown in Table 1, with 100 instances for each category, were considered.

The data used in the heuristic methods are described below:*L* (the size of the bar): 1188 mm;$l_{i}$ (length of item *i*): belongs to the range [20, 444] with uniform distribution;$d_{i}$ (demand of item *i*): according to [16], there are two cases: (i) low demand of items, $\left(d_{i}\in \left[1,3\right]\right)$ and (ii) high demand of items, $\left(d_{i}\in \left[4,15\right]\right)$ with uniform distribution.

As mentioned above, $d_{i}$ is based on [16], and *n* is generated uniformly in order to verify the heuristics performance, since in practice no example of this magnitude is known in the literature.

The Python language was used for all the heuristics implementation. The heuristic $RGR_{L1}$ [7] was used as the base to analyze the results, and the scipy library was used for the linear relaxation solution. The heuristic $RGR_{L1}$ was also adapted for the case considered in this study, in which there is no reuse of leftovers with stock; that is, the reuse parameter is defined as the smallest item to be produced in each instance. Still, in the heuristic $RGR_{L1}$, the initial cutting patterns (homogeneous), $c_{i}$, are generated from Equation (Equation 5).
(5)$$c_{i}=L-\left[\min\left(\left\lfloor \frac{L}{l_{i}}\right\rfloor ,d_{i}\right)\right]l_{i}$$

In *OptimizationTREE* and $RGR_{L1}$ heuristics, the sorting method used is quick sort. The hardware used for the simulations consists of an Intel processor^®^
$\;{\mathrm{Inside}}^{\mathrm{TM}}$ Core i7 920 @2.67 Ghz with 16 Gb of RAM. For the heuristic simulations, the Linux operating system Ubuntu 18.04 × 64 is used. The runtime of the presented instances is also considered in this paper and shows that the results are acceptable in practice.

## 5. Results and Discussion

The execution of the instances on each category provides results that allow the performance of the proposed heuristic to be analyzed in detail in comparison with the $RGR_{L1}$ heuristic, proposed by [7] and well known in the literature. The $RGR_{L1}$ heuristic considers a one-dimensional cutting stock problem in which the unused material in the cutting patterns may be used in the future if large enough. The authors of [7] modified some heuristics of the literature that minimize the trim loss and included the possibility of retails (large leftovers) that are not computed as losses.

It is important to highlight that the model proposed by [7] is the closest to the problem addressed in this paper. The main purpose of $RGR_{L1}$ is to solve the cutting stock problem with usable leftovers. The $RGR_{L1}$ algorithm was adapted to the restrictions considered in this study: only one type of bar in unlimited quantity and no stocks. It is also important to mention that other works in the literature that approach mathematical models with the use of leftovers have different objectives and would have to undergo more substantial adaptations, which would cause a mischaracterization of the problem.

Table 2 shows the mean of bars ($\bar{bar}$) used in the cutting process and its standard deviation σ(bar) as well as the mean of leftover ($\bar{left}$) and its standard deviation σ(left) of the *OptimizationTREE* and $RGR_{L1}$ heuristics.

Similarly, Table 3 shows the mean of loss ($\bar{loss}$) and its standard deviation σ(loss), the runtime for each category (in seconds) and its standard deviation for the *OptimizationTREE* and $RGR_{L1}$ heuristics.

In order to illustrate the performances of the heuristics, Figure 3 represents: (a) the means of cutting bars resulting from the *OptimizationTREE* and $RGR_{L1}$ heuristics, referring to the minimum number of bars (“B_bar”) and (b) the standard deviation of cutting bars. Figure 4 represents: (a) the means of leftovers, also calculated for the two heuristics, and mean of the leftover of the hypothetical cutting pattern and (b) the standard deviation of the leftover. Figure 5 shows the same for the losses, and Figure 6 presents the runtime for the heuristics under study.

These statistical measures indicate the performance across categories, considering the number of cutting bars, the leftover, the loss and the runtime of the *OptimizationTREE* heuristic and the heuristic proposed by [7]. Observing the results of Table 2 and Table 3, it can be observed that:The *OptimizationTREE* heuristic uses fewer bars than $RGR_{L1}$, presenting more leftovers and fewer losses.The *OptimizationTREE* heuristic has more leftovers and fewer losses when compared to the $RGR_{L1}$ heuristic.The $RGR_{L1}$ heuristic provides more losses, that is, small parts spread over the bars used in the cutting process. The *OptimizationTREE* heuristic provides fewer losses, which are concentrated in leftovers. Thus, this leftover can be used to address the next demand.Figure 6 shows that the runtime of the $RGR_{L1}$ heuristic is higher than that of the OptimizationTREE heuristic, especially for larger examples.Thus, although both heuristics provide a similar number of cutting bars, the $RGR_{L1}$ heuristic generates fewer leftovers and more losses than the heuristic proposed in this work, indicating worse performance.As expected, as shown in Figure 5, the *OptimizationTREE* heuristic generates the smallest number of loss, showing the more desirable performance.Figure 4 shows that the *OptimizationTREE* heuristic generates more leftovers. Generating leftovers is desirable when the number of cutting bars is close to $B_{bar}$, since leftovers are used to meet next demands. In the proposed heuristics, the leftovers are concentrated in the last bars. It is important to note that the peaks presented by the OptimizationTREE heuristic in Figure 4 are due to the use of the smallest items. In addition, the smallest item for each instance is fixed according to the smallest value of $l_{i}$. During the execution of the algorithm, the demand for the items is met, and then these items are no longer considered for cutting. The remaining items are larger items since the algorithm removes a unit from the largest item in each iteration. Therefore, from a given bar, more leftovers and very few losses are generated, since the higher demand causes these last cutting patterns to be repeated more often, causing the visible peaks.

Another analysis can be carried out between the hypothetical cutting pattern that presents the minimum number of bars, the leftover and the loss resulting from this hypothetical pattern. The distance of these values and the results of heuristics are presented in the following equations. The ${\bar{GAP}}_{bar}$, defined in Equation (Equation 6), presents the mean value of the distance (absolute error) between the lower bound of bars ($B_{bar_{p}}$), shown in Equation (Equation 3), and the results of the *OptimizationTREE* (OptT) and $RGR_{L1}$ heuristics. The “*H*” in the equation represents the chosen heuristic.
(6)$${\bar{GAP}}_{bar}=\frac{\sum_{p=1}^{P}\left|B_{bar_{p}}-H_{bar_{p}}\right|}{P}$$

Similarly, the ${\bar{GAP}}_{left}$ (Equation (Equation 7)) and the ${\bar{GAP}}_{loss}$ (Equation (Equation 8)) show the main value of the distance (absolute error) between lower bound of leftovers ($B_{left_{p}}$) and of loss ($B_{loss_{p}}$), and the heuristic result of the leftover ($H_{left_{p}}$) and of loss ($H_{loss_{p}}$), respectively, for each category.
(7)$${\bar{GAP}}_{left}=\frac{\sum_{p=1}^{P}\left|B_{left_{p}}-H_{left_{p}}\right|}{P}$$
(8)$${\bar{GAP}}_{loss}=\frac{\sum_{p=1}^{P}\left|B_{loss_{p}}-H_{loss_{p}}\right|}{P}$$

Table 4 and Table 5 show the values of the GAP obtained by the *OptimizationTREE* and $RGR_{L1}$ heuristics, respectively, considering the number of cutting bars, the leftover and the loss for each category. The columns present the numeric values provided by Equations (Equation 6), (Equation 7) and (Equation 8), respectively, followed by their respective standard deviations.

Analyzing Table 4 and Table 5, the *OptimizationTREE* heuristic presents the smallest GAP in relation to the number of cutting bars ($B_{bar}$). *OptimizationTREE* is closer to $B_{bar}$ and generates even larger leftovers.

The computational complexity of the proposed algorithms is shown in Table 6, considering the worst case of the two heuristics proposed and the heuristic $RGR_{L1}$ from the literature. The second column presents the number of sorts performed by each algorithm during its execution, and the last column shows the complexity order of the algorithms, considering the worst case.

Table 6 shows that the OptimizationTREE algorithm presents, in the worst case, a quadratic order computational complexity ($O(n^{2})$) but provides more leftovers to be reused, which is desirable, since *OptimizationTREE* aims to convert losses into leftovers, providing reuse. The computational complexity of the $RGR_{L1}$ heuristic is a polynomial order. The main runtime for all the 10 categories (in seconds) is as follows: *OptimizationTREE* = 0.991 and $RGR_{L1}$ = 3.4727. Then, for the case that considers unlimited quantity of one type of bar, OptimizationTREE is more appropriate than $RGR_{L1}$.

## 6. Conclusions

In this study, one-dimensional cutting stock problems were considered. These problems minimize the number of cutting bars in the cutting process, satisfying the demand, considering cutting rolls with known demand and using the leftovers provided from the previous roll’s cutting process. In order to accomplish this goal, the *OptimizationTREE* heuristic is proposed. It presents a tree-based structure and groups losses in the same bar to transform them into usable leftovers. The performance of the proposed heuristic was compared with the $RGR_{L1}$ heuristic, proposed by [7], to solve the problem considered in this paper (unlimited quantity of one type of bar and no stocks).

Analyzing Table 4 and Table 5, which show the values of the GAP obtained by *OptimizationTREE* and $RGR_{L1}$ heuristics, respectively, we can conclude that the *OptimizationTREE* heuristic presents the smallest GAP in relation to the number of cutting bars ($B_{bar}$). Therefore, *OptimizationTREE* is closer to the theoretical minimum limit, representing the fundamental contribution of this paper. In addition, the results show that the proposed heuristic reduces the processing time and the number of bars needed in the cutting process, while it provides a larger leftover for the one-dimensional cutting stock problem.

Using the *OptimizationTREE* heuristic proposed in this paper, the leftovers from previous cutting process can be used in the next item’s demands. The number of bars is minimized. Then, fewer losses are produced, and the losses are converted into leftovers. In this paper, a loss is an unused bar smaller than the size of the smallest item demanded. The loss can also be defined by the producer, for example as a fixed value or a minimum acceptable percentage. Therefore, the results offer a possible use for leftovers, reducing the use of raw materials in industrial processes and providing more commercially viable solutions.

Material loss is an important factor for the industry. In the production phase, the cutting process is present in several segments, such as in the production of tubes, profiles and door frames. The best use of materials provides a reduction in investments in production, for example, in actions such as the acquisition and transportation of raw materials and cost reduction for the end consumer, since the better use of materials in production makes it possible to reduce the final value for the consumer.

Finally, the proposed heuristics were implemented as open source (Python language), and the code is freely available in the GitHub package resource (*OptimizationTREE*: github.com/omatheuspimenta/heuristictree, accessed on 8 November 2023), which enables the use of the proposed heuristic in future works.

Inclusion of a third phase in the algorithms, i.e., after defining the all cutting patterns, changing the item distribution with the aim of converting losses in leftovers, is a prospect for continuing this research. This new phase must be carefully designed since it can increase the processing time.

## Figures and Tables

**Figure 1 materials-16-07133-f001:**
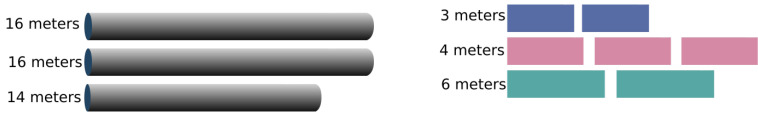
Available bars and demands

**Figure 2 materials-16-07133-f002:**
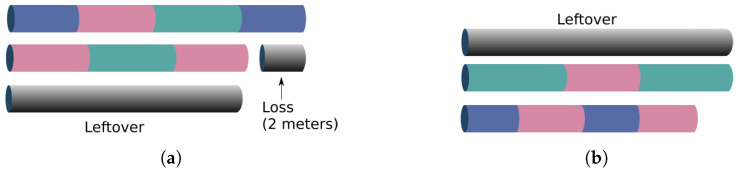
Steel coin cutting patterns: (**a**) Cutting pattern with loss and leftover; (**b**) Cutting pattern only with leftover.

**Figure 3 materials-16-07133-f003:**
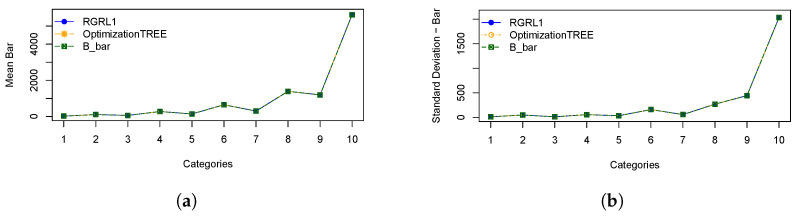
Statistical measures for the number of cut bars: (**a**) Mean of cut bars; (**b**) Standard deviation of cut bars.

**Figure 4 materials-16-07133-f004:**
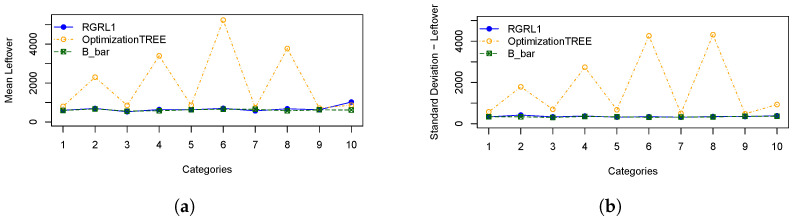
Statistical measures for the leftovers: (**a**) Mean of total leftover; (**b**) Standard deviation of total leftover.

**Figure 5 materials-16-07133-f005:**
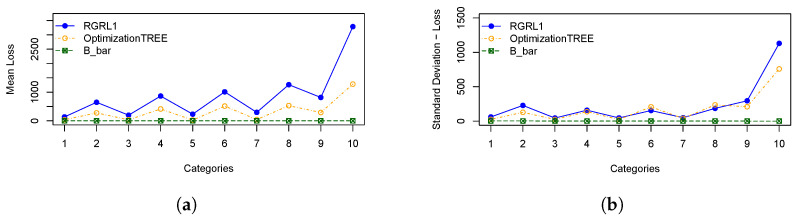
Statistical measures for the loss: (**a**) Mean of total loss; (**b**) Standard deviation of total loss.

**Figure 6 materials-16-07133-f006:**
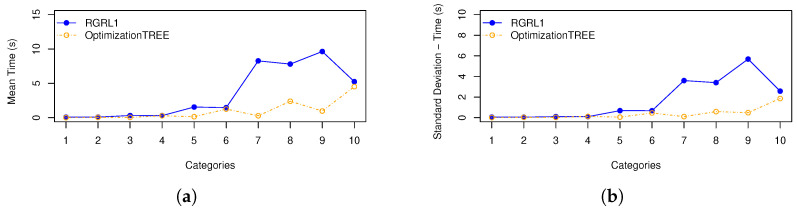
Statistical Measures for the runtime: (**a**) Mean of total runtime; (**b**) Standard deviation for the runtime.

**Table 1 materials-16-07133-t001:** Data categories.

**1.** n∈[5,100] **and** $d_{i}\in \left[1,3\right]$	**6.** n∈[201,500] **and** $d_{i}\in \left[4,15\right]$
**2. n∈[5,100] and $d_{i}\in \left[4,15\right]$**	**7. n∈[501,1000] and $d_{i}\in \left[1,3\right]$**
**3. n∈[101,200] and $d_{i}\in \left[1,3\right]$**	**8. n∈[501,1000] and $d_{i}\in \left[4,15\right]$**
**4. n∈[101,200] and $d_{i}\in \left[4,15\right]$**	**9. n∈[1001,5000] and $d_{i}\in \left[1,3\right]$**
**5. n∈[201,500] and $d_{i}\in \left[1,3\right]$**	**10. n∈[1001,5000] and $d_{i}\in \left[4,15\right]$**

**Table 2 materials-16-07133-t002:** Results of *OptimizationTREE* and $RGR_{L1}$ heuristics considering the mean of bars and the mean of leftover.

Category	*OptimizationTREE*				${\mathrm{RGR}}_{L1}$			
	$\bar{\mathrm{bar}}$	σ(bar)	$\bar{\mathrm{left}}$	σ(left)	*bar*	σ(bar)	$\bar{\mathrm{left}}$	σ(left)
1	20.99	11.78	799.56	574.7	21.88	11.66	599.61	332
2	107.82	48.59	2302.19	1788.5	107.72	48.09	693.39	420.94
3	59.87	12.13	851.16	699.83	60.7	12.14	525.83	333.6
4	277.13	55.97	3398.03	2737.4	276.11	55.63	637.62	373.86
5	140.73	34.19	859.94	674.7	141.68	34.34	625.08	318.15
6	651.67	158.08	5232.73	4258.91	649.22	158.71	699.69	342.34
7	298.98	57.25	727.03	502.05	300.02	57.32	570.61	315.35
8	1394.23	268.19	3773	4314.32	1393.22	269.28	676.49	347.14
9	1194.02	443.73	709.71	479.27	1195.36	443.85	628.24	346.26
10	5628.47	2035.61	862.02	931.35	5631.52	2036.55	1022.91	388.67

**Table 3 materials-16-07133-t003:** Results of *OptimizationTREE* and $RGR_{L1}$ heuristics considering the mean of loss and runtime.

Category	*OptimizationTREE*				${\mathrm{RGR}}_{L1}$			
	$\bar{\mathrm{loss}}$	σ(loss)	$\bar{\mathrm{time}}$	σ(time)	$\bar{\mathrm{loss}}$	σ(loss)	$\bar{\mathrm{time}}$	σ(time)
1	38.1	25.007	0.007	0.006	135.81	60.32	0.07	0.055
2	276.57	127.631	0.041	0.036	646.43	228.03	0.087	0.051
3	34.8	23.909	0.044	0.018	196.73	44.69	0.311	0.1
4	413.35	136.797	0.277	0.121	864.38	158.6	0.297	0.097
5	24.45	22.85	0.143	0.053	232.15	46.19	1.55	0.682
6	513.59	206.945	1.25	0.455	1010.63	153.5	1.46	0.674
7	42.66	39.605	0.27	0.104	294.67	49.8	8.27	3.6
8	528.87	235.843	2.39	0.589	1260.82	185.04	7.8	3.4
9	289.26	209.275	0.968	0.482	814.75	295.14	9.635	5.687
10	1281.6	758.561	4.52	1.86	3286.27	1129.68	5.247	2.574

**Table 4 materials-16-07133-t004:** GAP values obtained by *OptimizationTREE* heuristic.

Categories	*OptimizationTREE*		
	${\bar{\mathrm{GAP}}}_{\mathrm{bar}}\;\left(\pm \sigma \left({\mathrm{GAP}}_{\mathrm{bar}}\right)\right)$	${\bar{\mathrm{GAP}}}_{\mathrm{left}}\;\left(\pm \sigma \left({\mathrm{GAP}}_{\mathrm{left}}\right)\right)$	${\bar{\mathrm{GAP}}}_{\mathrm{loss}}\;\left(\pm \sigma \left({\mathrm{GAP}}_{\mathrm{loss}}\right)\right)$
**1**	0.23 (±0.48)	301.95 (±537.00)	37.59 (±25.24)
**2**	1.68 (±1.56)	1748.09 (±1743.92)	276.27 (±127.06)
**3**	0.31 (±0.61)	382.13 (±678.21)	34.69 (±23.83)
**4**	2.82 (±2.29)	2857.63 (±2725.46)	413.35 (±136.79)
**5**	0.24 (±0.56)	294.21 (±631.27)	24.37 (±22.82)
**6**	4.37 (±3.60)	4577.59 (±4306.75)	513.51 (±206.93)
**7**	0.14 (±0.42)	161.09 (±415.47)	42.66 (±39.60)
**8**	3.23 (±3.66)	3327.42 (±4220.75)	528.53 (±236.17)
**9**	0.35 (±0.53)	413.78 (±417.49)	289.26 (±209.27)
**10**	1.37 (±1.02)	581.53 (±817.98)	1281.6 (±758.56)

**Table 5 materials-16-07133-t005:** GAP values obtained by $RGR_{L1}$ heuristic.

Categories	${\mathrm{RGR}}_{L1}$		
	${\bar{\mathrm{GAP}}}_{\mathrm{bar}}\;\left(\pm \sigma \left({\mathrm{GAP}}_{\mathrm{bar}}\right)\right)$	${\bar{\mathrm{GAP}}}_{\mathrm{left}}\;\left(\pm \sigma \left({\mathrm{GAP}}_{\mathrm{left}}\right)\right)$	${\bar{\mathrm{GAP}}}_{\mathrm{loss}}\;\left(\pm \sigma \left({\mathrm{GAP}}_{\mathrm{loss}}\right)\right)$
**1**	1.12 (±0.32)	239.26 (±296.93)	135.3 (±60.06)
**2**	1.58 (±0.53)	525.71 (±245.00)	646.13 (±227.29)
**3**	1.14 (±0.34)	303.62 (±274.92)	196.62 (±44.67)
**4**	1.8 (±0.44)	431.62 (±288.11)	864.38 (±288.11)
**5**	1.19 (±0.39)	365.57 (±284.52)	232.07 (±46.27)
**6**	1.92 (±0.30)	226.49 (±239.94)	1010.55 (±153.50)
**7**	1.18 (±0.38)	393.75 (±227.98)	294.67 (±49.80)
**8**	2.22 (±0.43)	354.45 (±342.52)	1260.48 (±184.99)
**9**	1.69 (±0.48)	389.49 (±267.79)	814.75 (±293.66)
**10**	4.42 (±1.27)	523.2 (±342.21)	3286.27 (±1124.01)

**Table 6 materials-16-07133-t006:** Computational complexity.

Algorithm	Sorts	Order
*OptimizationTREE*	1	$O(n^{2})$
$RGR_{L1}$	2	$O(n^{k}),k\in N$

## Data Availability

Data are contained within the article.

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
