# Peer review of "A Tree-Based Heuristic for the One-Dimensional Cutting Stock Problem Optimization Using Leftovers"

_materials, 2023, doi:10.3390/ma16227133_

Round 1

Reviewer 1 Report

Comments and Suggestions for Authors

This paper presents a tree-based heuristic method for minimizing the number of cut bars in the one-dimensional cutting process, satisfying the items demand in an unlimited bars quantity of just one type. However, the paper still has some problems need to be improved. Thus, this paper should be major revised according to following comments:

  1. This paper proposes aheuristic algorithm and compares with the RGRL1 algorithm, but more description of the RGRL1 algorithm should be mentioned rather than just quoted.
  2. This paper only compares with RGRL1algorithm which proposed in 2009 and shows its effect in shortening processing time and reducing the number of bars, but it is uncertain whether it is novel compared with other new research.
  3. The technical depth of the paperis insufficient and the innovation point is not clear.
  4. In the “Results and Discussion”, the analysis of the results is too complicated and has many repetitions, which should be simplified.
  5. There are some details in the article that need to be corrected, such as there is no Equation (9), Figure 7-9.
  6. The format of figures and tables needs to be improved.
  7. In the conclusion part, the summary of the content of the paper is too simple and it is not clear about the optimization effect. The two analysis methods can be summarized respectively.
  8. The actual positive impact of the heuristic on industrial processes is not analyzed in the text, but only mentioned in the abstract.

Author Response

Response to Reviewer 1 Comments

The response to Reviewer 1 comments is described point by point as follows. Text changes are highlighted in blue.

The authors are grateful for the reviewer’s suggestion that greatly contributed to the improvement of the text.

  1. This paper proposes a heuristic algorithm and compares with the RGRL1 algorithm, but more description of the RGRL1 algorithm should be mentioned rather than just quoted.

A brief description of the RGRL1 algorithm was included in the first paragraph of Section 5 in order to provide more details.

2. This paper only compares with RGRL1algorithm which proposed in 2009 and shows its effect in shortening processing time and reducing the number of bars, but it is uncertain whether it is novel compared with other new research.

The model proposed by [1] is the closest to the problem addressed in this paper. The main purpose of RGRL1 is to solve the cutting stock problem with  usable leftover. Then, RGRL1 algorithm was adapted to the restrictions considered in this paper: only one type of bar in unlimited quantity and no stocks. It is important to mention that other works in the literature that approach mathematical models with the use of leftovers would have to undergo more substantial adaptations, which would cause a mischaracterization of the problem. This information was included in the second paragraph of Section 5.

3. The technical depth of the paper is insufficient and the innovation point is not clear.

The contributions of the paper are described in the penultimate paragraph of the introduction. In the revised version of the paper, the conclusion section has been improved in order to clarify this question.

4. In the “Results and Discussion”, the analysis of the results is too complicated and has many repetitions, which should be simplified.

In the “Results and Discussion” section of the new version of the paper, tables and analysis have been simplified in order to answer this question.

5. There are some details in the article that need to be corrected, such as there is no Equation (9), Figure 7-9.

The numbering of figures and tables has been corrected in the new version of the article

6. The format of figures and tables needs to be improved.

The new version of the paper brings a better organization of the tables and their explanation in Section 5.

7. In the conclusion part, the summary of the content of the paper is too simple and it is not clear about the optimization effect. The two analysis methods can be summarized respectively.

New information has been added in the Conclusion section in order to highlight the contributions and th optimization effect of the results.

8. The actual positive impact of the heuristic on industrial processes is not analyzed in the text, but only mentioned in the abstract.

The Conclusion section has been reformulated to make clear the contributions and the positive impact of the proposed heuristic.

Reviewer 2 Report

Comments and Suggestions for Authors

Considering the cutting stock problems and usable leftovers, this paper presents a tree-based heuristic method, denominated OptimizationTREE, for minimizing the number of cut bars in the one-dimensional cutting process, satisfying the items demand in an unlimited bars quantity of just one type. Performances of the proposed heuristic method are analyzed and compared with RGRL1 algorithm. The heuristic will help to reduce the use of raw materials and promote the development of green manufacturing. The research work presented in the manuscript is scientifically and technologically justifiable. But there are some questions:

1、Spaces between numbers and units were absent, e.g. 2.67Ghz and 16Gb in page 7. Please check the problems throughout in the manuscript, carefully.

2、Some minor errors exist in the manuscript, e.g. “Equation (9)” in page 7 and the figure number in page 9. Please check the problems throughout in the manuscript, carefully.

3、Some units have problems, e.g. “sec” in Figure 4. “sec” is the abbreviation of “second”, and the correct unit is “s”.

4、In Figure 2, “Standard Deviation of Cut Bars” may be “Standard Deviation of Total Leftover”.

5、In the research background and significance of this article, there should be more introduction about green manufacturing. The following papers describes this aspect, and the author could refer to them:

①H.M. Alswat, P.T. Mativenga. The International Dimension of Electrical Energy Derived Emissions for Machine Tools. Procedia CIRP, 2021; 98: 696-701.

②J.L. Wang, Y.B. Tian, X.T. Hu, Y. Li, K. Zhang, Y.H. Liu. Predictive modelling and Pareto optimization for energy efficient grinding based on aANN-embedded NSGA II algorithm. Journal of Cleaner Production, 2021; 327: 129479.

Author Response

Response to Reviewer 2 Comments

The response to Reviewer 2 comments is described point by point as follows. Text changes are highlighted in blue.

The authors are grateful for the recommendations that greatly contributed to the improvement of the text. In the new version of the paper, questions 1-4 have been promptly answered and the text has been corrected.

About question 5, the two references mentioned by the reviewer were cited in the Introduction section. However, the authors consider that a deeper discussion on green manufacturing would be out of the scope of the work, since the environment preservation is only one of the contributions of the work, as a consequence of the economy of raw materials.

Reviewer 3 Report

Comments and Suggestions for Authors

This paper presents a tree-based heuristic method for minimizing the number of cut bars in the one-dimensional cutting process, satisfying the items demand in an unlimited bars quantity of one type.
The results show that proposed heuristic reduces processing time and the number of bars needed in cutting process.
The authors write: "The cutting process can generate leftover (which can be reused in a new demand) or losses (which are discarded)". In my opinion, the authors should describe in more detail the cutting process from the technical and technological side.
The methodology used in the article is correct.

Author Response

Response to Reviewer 3 Comments

The response to Reviewer 3 comments is described point by point as follows. Text changes are highlighted in blue.

The authors are grateful for the reviewer’s suggestion that greatly contributed to the improvement of the text. A better description of the addressed problem has been included in Introduction section in order to clarify this question and to illustrate a cutting problem type that the proposed heuristic can solve.